# Intelligent Gas Sensors for Food Safety and Quality Monitoring: Advances, Applications, and Future Directions

**DOI:** 10.3390/foods14152706

**Published:** 2025-08-01

**Authors:** Heera Jayan, Ruiyun Zhou, Chanjun Sun, Chen Wang, Limei Yin, Xiaobo Zou, Zhiming Guo

**Affiliations:** 1China Light Industry Key Laboratory of Food Intelligent Detection & Processing, School of Food and Biological Engineering, Jiangsu University, Zhenjiang 212013, China; heerajayan93@outlook.com (H.J.); ryzhou@ujs.edu.cn (R.Z.); chanjun.sun@ujs.edu.cn (C.S.); wangchen@ujs.edu.cn (C.W.); yinlimei@ujs.edu.cn (L.Y.); zou_xiaobo@ujs.edu.cn (X.Z.); 2Belt-and-Road Joint Laboratory on Smart Agriculture Technology and Equipment, School of Integrated Circuits, Jiangsu University, Zhenjiang 212013, China; 3International Joint Research Laboratory of Intelligent Agriculture and Agri-Products Processing, Jiangsu University, Zhenjiang 212013, China

**Keywords:** gas sensor, food quality, VOCs, food spoilage, data analysis

## Abstract

Gas sensors are considered a highly effective non-destructive technique for monitoring the quality and safety of food materials. These intelligent sensors can detect volatile profiles emitted by food products, providing valuable information on the changes occurring within the food. Gas sensors have garnered significant interest for their numerous advantages in the development of food safety monitoring systems. The adaptable characteristics of gas sensors make them ideal for integration into production lines, while the flexibility of certain sensor types allows for incorporation into packaging materials. Various types of gas sensors have been developed for their distinct properties and are utilized in a wide range of applications. Metal-oxide semiconductors and optical sensors are widely studied for their potential use as gas sensors in food quality assessments due to their ability to provide visual indicators to consumers. The advancement of new nanomaterials and their integration with advanced data acquisition techniques is expected to enhance the performance and utility of sensors in sustainable practices within the food supply chain.

## 1. Introduction

Food safety and quality assurance have become critical concerns due to the continuous rise in food spoilage and contamination, leading to health risks and economic losses. The Food Waste Index Report 2024 released by the UN Environment Programme indicates that approximately 1.05 billion metric tons of food are wasted globally [1]. The World Health Organization has recognized that this persistent problem has significant implications for the global economy, as well as indirect consequences on climate change, biodiversity loss, and pollution. Additionally, the risk of foodborne illnesses due to food spoilage and contamination poses a serious threat, as evidenced by the 600 million incidents reported each year. Considering the increasing demand for safe and fresh food products by a well-informed consumer base, the development of advanced sensing technologies that enable continuous, fast, and accurate monitoring of food products across the supply chain is necessary [2].

Conventional methods used to assess the safety and quality of food products provide accurate assessments, but often have limitations such as high cost, high time consumption, and a lack of on-site applicability. These methods typically necessitate the expertise of a trained individual for operation, rendering them more appropriate for use in a laboratory setting rather than on-site monitoring. Additionally, the majority of the techniques, while standardized, do not provide preventative and proactive outputs. These challenges indicate the need to create more selective, cost-effective, user-friendly monitoring technologies specifically for food quality monitoring applications [3].

Gas sensing methods have emerged as a popular non-destructive method to assess the safety and quality of food products. Gas sensors detect volatile organic compounds (VOCs) released from food products, providing an understanding of the aroma and flavor profile of the product. Since any change in the composition of food causes a change in the volatile profile of the food product, monitoring the gases associated with food is a good way to understand food quality and safety [4]. For example, the ethylene concentration released during climacteric fruit is a good indicator of the maturity and ripeness of the food product, while ammonia production is a good indicator of spoilage in meat and dairy products. A highly sensitive sensor can detect the gases associated with food and can incorporate several different gas sensors to produce a gas sensor system that can detect multiple gases, providing insight into the physiochemical characteristics of the food. The early detection of gases ensures the timely prevention and control of food spoilage and quality deterioration. Gas sensors, with their high sensitivity, portability, and fast response, enable real-time monitoring of food throughout processing, storage, and transportation, unlike traditional methods. Further gas sensors can be integrated into smart packaging to continuously monitor and alert consumers to the spoilage and unsafe conditions of the food [5].

This review outlines the significance of gas sensor technologies in the realm of food safety and quality monitoring, emphasizing their role in ensuring that food products meet safety standards and consumer expectations. This discussion encompass various types of gas sensors, including metal-oxide semiconductors (MOS), electrochemical sensors, optical sensors, conducting polymer sensors, and sensor arrays, highlighting their principles of operation, advantages, and specific applications in food monitoring. Each sensor type presents unique capabilities that cater to different aspects of food safety, such as detecting spoilage gases, assessing freshness, and identifying adulterants. An exploration of the future directions of gas sensor technology, considering the existing challenges such as selectivity, sensitivity, and integration into packaging systems, is discussed in detail.

This review aims to provide a comprehensive overview of recent advances in gas sensor technologies for food safety and quality monitoring. Advances in gas sensing technologies offer promising solutions that incorporate current research trends in smart packaging, automation, and Internet of Things (IoT)-based food monitoring. By summarizing recent developments and outlining future opportunities, this review serves as a valuable resource for researchers and policymakers aiming to ensure food quality and reduce food waste.

## 2. Gas Sensor Principles and Types

Comparative overview of gas-sensing technologies for food applications, highlighting their key characteristics, advantages, limitations, and manufacturing methods, given in Table 1.

### 2.1. Metal-Oxide Semiconductor (MOS) Sensors

MOS sensors are the most widely used sensors for gas composition analysis in food safety and quality evaluation due to their low production cost and enhanced sensitivity. The surface of the semiconductor sensor interacts with gas molecules, resulting in a change in electrical conductivity (Figure 1A) [6]. Generally, oxygen molecules form a depletion layer that raises resistance by adhering to the sensor’s surface and removing electrons from the conduction band. Furthermore, when reducing gases such as hydrogen, methane, or carbon monoxide are present, electrons are released back to into the conduction band, resulting in a reduction in resistance. In contrast, the presence of oxidizing gases increase the resistance. The change in resistance is correlated with the gas concentration in the working environment. Based on the sensing mechanism, the MOS sensors are classified into N-type (resistance decreases in reducing gases) or P-type (resistance increases in reducing gas). Typically, tin dioxide, tungsten oxide, indium oxide, and zinc oxide are employed as a sensing material in MOS sensors, selected based on their chemical stability and oxygen absorption capacity [7]. Early MOS sensors employed traditional metal-oxides like SnO_2_; however, recent developments have involve composite and mixed metal-oxides, doping with noble metals, and nanostructures. A major advancement for MOS sensors has been in increasing their selectivity through innovative sensor designs.

MOS sensors have been shown to detect VOCs and spoilage gasses at extremely low concentrations (typically below ppm levels), allowing for the early and precise identification of changes in food quality. Their rapid detection and quick recovery time makes them suitable for development of real-time monitoring systems. MOS sensors are capable of detecting the wide range of gases released during food spoilage and contamination, such as NH_3_, CO_2_, H_2_S, and various other VOCs, especially in protein-rich foods like meat, seafood, and milk. This enable real-time monitoring of the quality and safety of food products throughout processing, storage, and transportation.

### 2.2. Electrochemical Sensors

Electrochemical sensors are widely used in gas monitoring for their reliable operation in detecting indicator gases during food spoilage [8]. In electrochemical gas sensors, the target gas is oxidized or reduced at the electrode, which produce electrical current proportional to the gas concentration. These sensors work using various mechanisms based on the parameters being measured: amperometric techniques quantify changes in current, potentiometric methods assess variations in potential, chemoresistive approaches monitor changes in resistance, and capacitive strategies analyze changes in capacitance (Figure 1B) [9]. The electrodes are modified and enhanced through the addition of materials in order to improve their electrical properties and enhance the specific capture of certain gas molecules. For example, modification with nanofibers offers high surface area and flexibility, resulting in improved sensor sensitivity and selectivity. Materials like graphene and carbon nanotubes enhances the electrochemical response, increasing accuracy and detection time [10]. Furthermore, affinity reagents such as antibodies, aptamers, and molecularly imprinted polymers are also employed to achieve the very specific detection of target gas molecules. The evolution of electrochemical sensors from bulky, single-target devices to miniaturized, low-power sensors has ensured their sensitivity and stability in measurements. Microfabrication techniques enabled their integration into portable systems for real-time analysis.

The most common electrochemical sensors commonly employed in the food sector are ethylene sensors, employed for monitoring fruit ripeness. These sensors use a working counter, often a reference electrode on an insulating substrate. The ethylene gas dissolves in an electrolyte and undergoes electrochemical oxidation at the working electrode, resulting in generating a current propositional to the ethylene concentration [11].

A major development in electrochemical sensors is the introduction of solid-state ion-conductive material, which involves replacing the liquid electrolyte, enabling the miniaturization of electrochemical sensors into chips. This shift in development enables the mass production of chip-based sensors with low energy consumption and long lifespans [12]. In addition, micro-electro-mechanical systems (MEMS) technology contributes to the improvement of electrochemical sensors through the precise manufacturing of miniature electrodes and the incorporation of other sensing elements onto a single platform [13].

### 2.3. Optical Sensors

Optical gas sensors have gained popularity in recent years due to their ability to provide visual results, making them widely researched in the field of gas detection technology. It is advantageous for consumers to have direct access to information regarding the quality and safety of food products without any specialized equipment or supplementary resources [14]. Optical sensors are categorized into colorimetric sensors and fluorescence sensors based on the mechanisms of action.

#### 2.3.1. Colorimetric

Colorimetric gas sensors detect gas concentrations by inducing a color change in response to the presence of the target gas, allowing for visual observation of sensor signals. The sensors operate based on the chemo-responsive dye that are capable of interacting with the target gases through specific reactive centers [15]. The chromophore component of the dye undergoes a color change in response to this interaction. The colorimetric sensor generate a unique pattern for the gas present, enabling the detection and differentiation of complex components, as shown in Figure 1C [16]. The chemo-responsive dye is selected based on the absorption coefficient and/or emission quantum yields for various applications. The chemo responsive dye is immobilized on the solid support, such as silica gel, organic polymers, and filter papers, via absorption entrapment covalent bonding [17]. The incorporation of metal nanoparticles has been evidenced to improve the performance of the sensor with a faster response time through the plasmonic effect. Furthermore, the development of innovative responsive dyes and hybrid materials improve sensor stability and selectivity. A multiplex approach utilizing a colorimetric sensor array was developed to mimic the olfactory system [18]. This approach generates specific fingerprints for each spoilage gas present in a mixture, allowing for the simultaneous detection of multiple spoilage indicative gases. Colorimetric sensors represent a straightforward technique in theory, yet the scientific advancement of the incorporation of novel chemo-responsive materials and their compatibility with IoT analysis renders them a promising choice for the creation of monitoring systems.

#### 2.3.2. Fluorescence Sensor

Fluorescence sensors employ fluorescent dyes or probes that emit light upon excitation by a specific wavelength, and the interaction with the gases result in changes in fluorescence. The change in fluorescence intensity is directly correlated to the concentration of target gas in the environment. A simple fluorescence-based sensing system monitors changes in fluorescence intensity at one specific wavelength, making it susceptible to environmental factors [19]. Ratiometric fluorescence sensors employ a built-in correction for variations by measuring analyte-induced changes in two or more emissions simultaneously [20]. For instance, the pyrene-based fluorescent sensor (PN) initially emits blue light at a wavelength of 446 nm through pyrene excimer formation before reacting with formaldehyde (FA). Later, a chemical rearrangement takes place, resulting in the formation of a new compound, PN-FA, which emits violet light at a wavelength of 390 nm. The sensor enhances signal accuracy and stability by quantitatively comparing the intensity ratio of two emissions, thus reducing the impact of environmental and instrumental fluctuations [21].

### 2.4. Conducting Polymer Sensors

Conducting polymer sensors are developed by interspersing conducting polymers such as polyaniline, polypyrrole, polythiophene, and their derivatives into an insulating matrix, creating a chemoresistive and electrochemical transduction system [22]. The absorption of gas induces changes in electrical conductivity. The absorption of gas is often a reversible interaction, enabling multiple and real-time monitoring. Unlike MOS sensors, conducting polymer sensors operates efficiently at ambient conditions. One significant benefit of conducting polymer sensors is their high flexibility, allowing for their easy integration into packaging materials. These advantages render them valuable assets for scientific research, particularly in enhancing a sensor’s sensitivity. One approach involves the synthesis of nanostructures, including spheres, wires, rods, and tubes, within a conductive polymer matrix [23]. Another approach is the physical blending of conductive nanomaterials, such as carbon nanotubes and graphene, with polymers. The incorporation of a metal–organic framework (MOF) into the conductive polymer (3,4-propylenedioxythiophene) enhances the intrinsic properties of both polymers and MOFs, resulting in chemiresistive sensors with higher sensitivity, selectivity, and faster recovery for detecting NO_2_ gas [24]. Similarly, the gas-sensing properties of conducting polymer sensors have shown improvement when integrated with nanofibers, nanotubes, and networks that facilitate gas diffusivity. In the utilization of conductive polymers for sensor applications, 3D printing has emerged as a manufacturing technique that allows for customization of the sensor surface and porosity to optimize performance for specific applications. Various printing techniques, such as inkjet, extrusion, and light-based printing, enable the efficient and scalable production of disposable or commercial food sensors [25]. Recent research has integrated conducting polymers into portable analytical platforms, enabling the on-site monitoring of food spoilage and freshness [26]. Flexible, wearable formats and their integration with wireless systems are major advancements that enable wide applications.

### 2.5. Sensor Array

An electronic nose is an intelligent analytical device developed to evaluate volatile compounds using a simulation of the human olfactory system. This is achieved by employing a group of sensors that can interact with the molecules generated from food products, resulting in the generation of a fingerprint for each sample. Electronic noses have been developed as a solution to eliminate bias in traditional human sensory evaluations, in response to market demand, as depicted in Figure 1D [27]. The use of electronic noses is gaining popularity due to their versatility throughout the entire food supply chain, from production to consumption. These advancements have shifted sensors array from basic multi-sensor setups to advanced electronic noses incorporating machine learning and pattern recognition algorithms [28].

Both electronic noses and tongues work using a similar principle with arrays of sensors. Electronic noses consist of gas sensor arrays, while electronic tongues consist of chemical sensor arrays. The setup of both systems is extremely similar, including sensors, signal collectors, and a computer system [29,30]. The sensors detect the molecules and record the response signal generated, the signal collector realizes the transmission and preprocessing of these signals, and the computer system provides comprehensive judgment of the data by employing appropriate algorithms.

The electronic nose is a gas sensor array that responds to specific volatiles in food. The commonly employed sensors for constructing an electronic nose include conducting polymers (CPs), quartz crystal microbalance (QCM), metal-oxide semiconductor (MOS), and surface acoustic wave (SAW) sensors [31]. SAZ and QCM sensors detect changes in mass on the surface caused by the adsorption of volatile compounds, which, in turn, alter acoustic wave propagation or resonant frequency. These are less commonly used for sensor array development, but rather serve a specific purpose. In order to improve reliability and accuracy, an e-nose sensor should be equipped with 5–30 sensors, often with multiple replicates of each sensor. These arrays utilize the partial specificity and interference reactivity of each sensor to address complex volatile mixtures. The design of the e-nose system varies depending on the specific application. Commercial e-nose systems are not capable of detecting all of the volatile components in food. Therefore, significant efforts are made towards tailoring the sensor based on specific needs.

Improvements in existing gas sensor technology, along with the incorporation of gas sensors into packaging materials and the creation of multisensory sensor systems, have proven to be effective in analyzing spoilage and monitoring freshness in various food products. Integrating gas sensor data with deep learning techniques is paving the way for the advancement of intelligent monitoring systems.

**Figure 1 foods-14-02706-f001:**
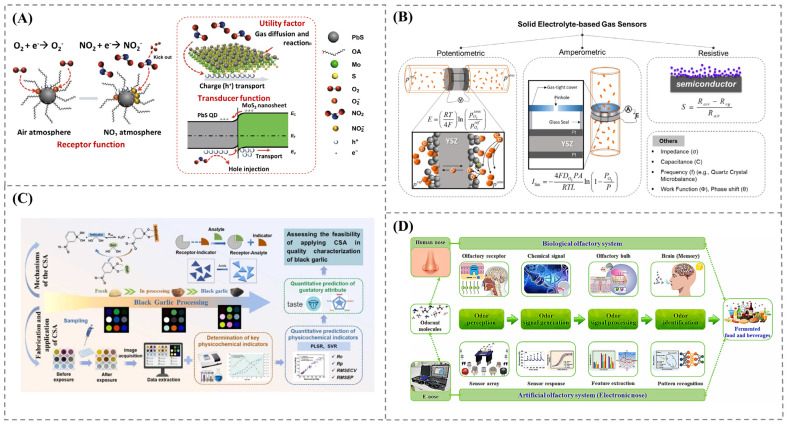
(**A**) Metal-oxide semiconductor gas sensor [6]; (**B**) different types of electrochemical gas sensors [9]; (**C**) colorimetric gas sensor for quality evaluation of black garlic [16]; (**D**) e-nose mechanism in comparison to the human olfactory system [27].

## 3. Applications in Food Quality and Safety

This section provides an overview of the advanced uses of gas sensors in a variety of applications related to food safety and quality analysis. A summary of the recent works involving gas sensors is discussed in Table 2.

### 3.1. Spoilage and Freshness Monitoring

Microbial activity and chemical reactions contribute to the release of volatile gases from food products during spoilage or decomposition. Monitoring the levels of specific gases is a reliable method for assessing the freshness of particular food products and detecting spillage at an early stage. Foods high in protein, such as seafood and meat, are sources of volatile nitrous compounds produced through enzymatic degradation of proteins and amino acids. The primary compounds present in seafood spoilage are ammonia, trimethylamine, and dimethylamine [32]. Total Volatile Basic Nitrogen (TVB-N) is commonly utilized as a marker for seafood spoilage, although it is more effective in detecting the advanced stages of spoilage rather than in early monitoring systems. Other common gaseous spoilage indicators are hydrogen sulfide, methane, ethylene, and nitrogen dioxide.

Colorimetric sensing has gained significant attention due to offering real-time and visual indications of food freshness. A study using anthocyanin integrated chitin whiskers-gelatin films demonstrated a high level of efficacy due to its straightforward methodology and ability to visually detect spoilage in pork through pH-induced color change [33]. While this method provides a direct means of integration into packaging materials, its sensitivity is constrained to broad pH changes and lacks specificity for particular spoilage gases. In contrary, a cellulose acetate/MOF-based sensor developed by Hashemian et al. [34] offers high specific detection of ammonia produced during meat spoilage with a very low detection limit of 0.02 ppm. The incorporation of a smartphone application enhances the versatility of the platform, making it suitable for the development of real-time monitoring systems. A further advancement in colorimetric analysis was proposed by Lin et al. [35], where a CSA was developed for detecting trimethylamine (TMA) at ppb levels, as shown in Figure 2A. Combining sensor data with a deep learning algorithm resulted in the precise prediction of the freshness of chilled beef. Unlike a single-indicator system, the multichannel input array of CSA was able to differentiate various gas profiles, even with subtle changes in concentration. Research on colorimetric sensing has aimed to include a variety of target analytes and foods through various technological integrations, reflecting its progression from simple indicators to intelligent diagnostic tools.

MOS sensors are found to be highly effective for detecting gases associated with spoilage, particularly ammonia and trimethylamine. The (001) TiO_2_/Ti_3_C_2_Tx heterostructure sensor demonstrated enhanced sensitivity to ammonia, achieving ultra-low detection limits, enhanced by UV illumination, crucial for assessing freshness of fish, pork, and shrimp [36] (Figure 2B). The NiCo_2_O_4_-ZnO composite sensor shows significant improvements in response and recovery times for trimethylamine detection [37]. This sensor works based on the resistance change during gas adsorption and desorption. Whereas the development of the (001) TiO_2_/Ti_3_C_2_Tx and NiCo_2_O_4_-ZnO sensor was primarily aimed to improve the sensitivity and selectivity of the gas sensor for specific gases with different material compositions, Huang et al. [38] developed a comprehensive approach through the combination of gas, temperature, and impedance measurement, as shown in Figure 2C. The flexible sensing system specifically designed for assessing lamb freshness was capable of adapting to a packaging material for continues real-time monitoring of freshness.

The ability of an e-nose to mimic human odor perception was utilized to analyze the flavor profile of a chicken drumstick during different sugar smoking times, revealing differences in odor profiles among the samples and identifying the smoked samples with the strongest odor characteristics [39]. The combination of the e-nose with machine vision and chemometric methods showed high classification accuracy in detecting spinach freshness (up to 93.75%) and identifying spoilage fungi in apples (over 93.06%) [40,41].

The current research emphasis has transitioned towards expanding the capabilities of gas sensors, particularly related to multisensory data fusion to improve accuracy and sensitivity in the monitoring of freshness and spoilage [42]. Li et al. [43] addressed the limitations of hyperspectral imaging (HSI) in detecting aromatic compounds by integrating HSI with an e-nose and olfactory visualization system. The fusion approach achieved a classification accuracy of 92%, demonstrating the complementary behavior of gas sensor data with visual technologies for discrimination between different grades of green tea. Similarly, Ren et al. [44] employed a smart multisensory device that combined olfactory, taste, and visual sensors, achieving high sensitivity in both maturity detection (95%) and brand classification (91.67%) of preserved eggs. This strategy proved the effectiveness of sensor integration in monitoring subtle chemical and sensory changes in food over time. The multisensory data fusion system can provide a more holistic approach to quality evaluation of tomatoes [45]. The e-nose detected volatile gases emitted during ripening, while computer vision system captured the color, providing a more detailed understanding of ripeness and hardness.

**Figure 2 foods-14-02706-f002:**
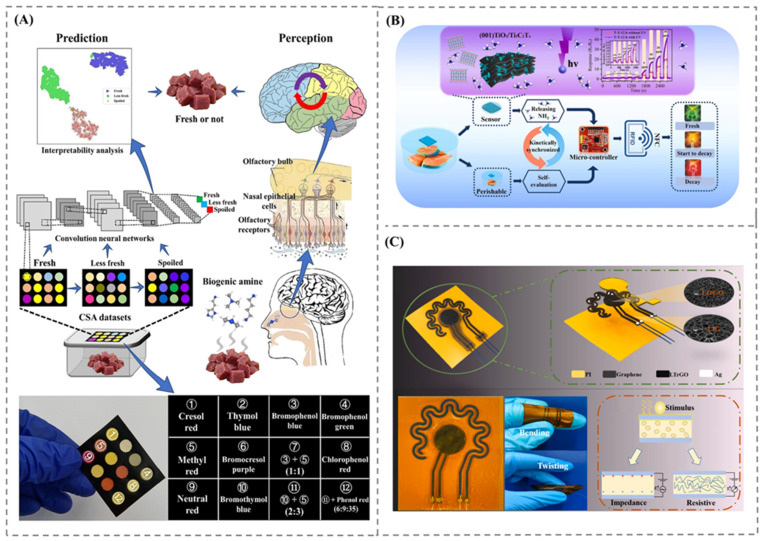
(**A**) Colorimetric sensor array for detecting trimethylamine for freshness analysis of chilled beef. Adapted with permission from Ref. [35]. Copyright 2023 Elsevier. (**B**) (001) TiO_2_/Ti_3_C_2_Tx heterostructure sensor for detecting ammonia for freshness of meat, Adapted with permission from Ref. [36]. Copyright 2022 Elsevier. (**C**) flexible sensing system specifically designed for assessing lamb freshness, Adapted with permission from Ref. [38]. Copyright 2023 Elsevier.

### 3.2. Authenticity and Adulteration Detection

The use of gas sensors, particularly electronic noses, has been widely utilized for successful non-destructive detection of adulteration. However, an increasing body of research shows that performance and adaptability vary based on food composition, sensor design, and data processing methods. For example, adulteration of sesame oil was detected using a nine-sensor MOS electronic nose coupled with gas chromatography-mass spectrometry (GC-MS) with a high accuracy of 98.7%, demonstrating the effectiveness of integrating sensor data with traditional chemical analysis [46]. In contrast, adulteration of powdered milk with whey was detected using an eight-sensor electronic nose, resulting in slightly lower accuracy (95.6%), despite employing similar data-processing methods such as principal component analysis (PCA) and artificial neural networks (ANNs) [47]. This may be due to the challenge of detecting non-volatile adulterants in a complex matrix like milk. Furthermore, the utilization of the BME688 gas sensor matrix in detecting olive oil adulteration demonstrated exceptionally high precision through the application of advanced machine learning algorithms [48]. The developed sensor features compact and cost-effective hardware that is highly sensitive to VOCs, as shown in Figure 3A. Meanwhile, a portable e-nose sensor developed for detecting soybean protein isolate in minced chicken demonstrated strong classification accuracy, paving the way for on-site adulteration detection in perishable food products [49]. When compared to other food products, meat has a high level of microbial contamination, which presents additional challenges. This makes this portable sensor notable for real-time analysis methods. The portability of the electronic nose is enhanced by utilizing a robust sensor array, integration with machine learning, and a compact design. These advancements were utilized by Putri et al. [50] to create a portable electronic nose for classifying different types of meat floss, specifically beef, chicken, and pork (Figure 3B). In addition to using lightweight materials and battery power, the design also incorporates the time window slicing method during feature extraction, leading to quick assessment in portable applications.

Technological advancements in gas sensing methods have led to the precise classification of food products based on their geographical origin by detecting subtle differences in volatile compounds. Zeng et al. [51] developed a comprehensive approach combining an electronic nose, GC-MS, and multivariate analysis to successfully classify *Zanthoxylum bungeanum Maxim* (ZBM) from Sichuan from other regions. Similarly, the strength of coupling advanced sensing methods with machine learning to decode complex volatile signatures was employed by Han et al. [52] for red wine authentication. The integration of a low-cost electronic nose and voltametric electronic tongue enabled classification based on geographical origin, grape variety, and brand. The integration of olfactory and gustatory sensing, combined with a machine learning model, provides a multidimensional analytical approach to handle complex foods in which both aroma and taste markers contribute to variability. Building on these sensitive high-performing approaches, Arslan et al. [53] developed a portable smartphone-based colorimetric sensor array in order to classify rice cultivars by origin. Despite being a simplified approach, the method achieved 100% classification accuracy based on the color difference maps generated in response to volatile compounds from rice. Its low cost and real-time applicability position it as a promising tool for on-site authenticity testing.

The integration of an e-nose with other complementary techniques, such as an e-tongue and physiochemical assays, enhances the information obtained from the sample, broadening the understanding of complex matrices [54]. The combination of an e-nose and an e-tongue has been shown to enhance the predictive modeling of sensory attributes in fermented foods, outperforming models based on single-sensor output [55]. This integrated approach has the capability to capture both volatile and non-volatile compounds, providing a deeper profile of fermented soybean paste. Furthermore, data fusion using the mid-level fusion method increased the discrimination accuracy to 97.22% for various soybean paste samples [56]. The approach provides a comprehensive flavor profile that differentiates foods based on differences in sensory clusters. However, the approach has the potential to serve as an alternative to a human sensory panel. Yu et al. [56] identified 57 VOCs, primarily esters and alcohols, using HS-SPME-GC/MS, reinforcing the efficacy of integrated systems in understanding the olfactory components associated with consumer perception. A strong correlation was observed between e-tongue readings and the physicochemical properties associated with taste, suggesting the importance of these tools in quality control. The utilization of an e-nose for the analysis of the flavor profile of Dushan shrimp paste identified the presence of distinct volatile compounds, including inorganic sulfur compounds and nitrogen oxides, produced by specific bacterial species during the fermentation process (Figure 3C). These compounds were found to be associated with physicochemical alterations, such as changes in pH, moisture content, and color [57].

The e-nose demonstrates significant utility in the realm of food authenticity and adulteration detection. When combined with complementary techniques, the e-nose can provide a more comprehensive and extensive analysis of food products. In such cases, data-fusion techniques are utilized to extract reliable information from data, thereby enhancing food authentication and quality control processes. The portability of electronic nose devices is a crucial factor in enabling on-site detection of food adulteration.

**Figure 3 foods-14-02706-f003:**
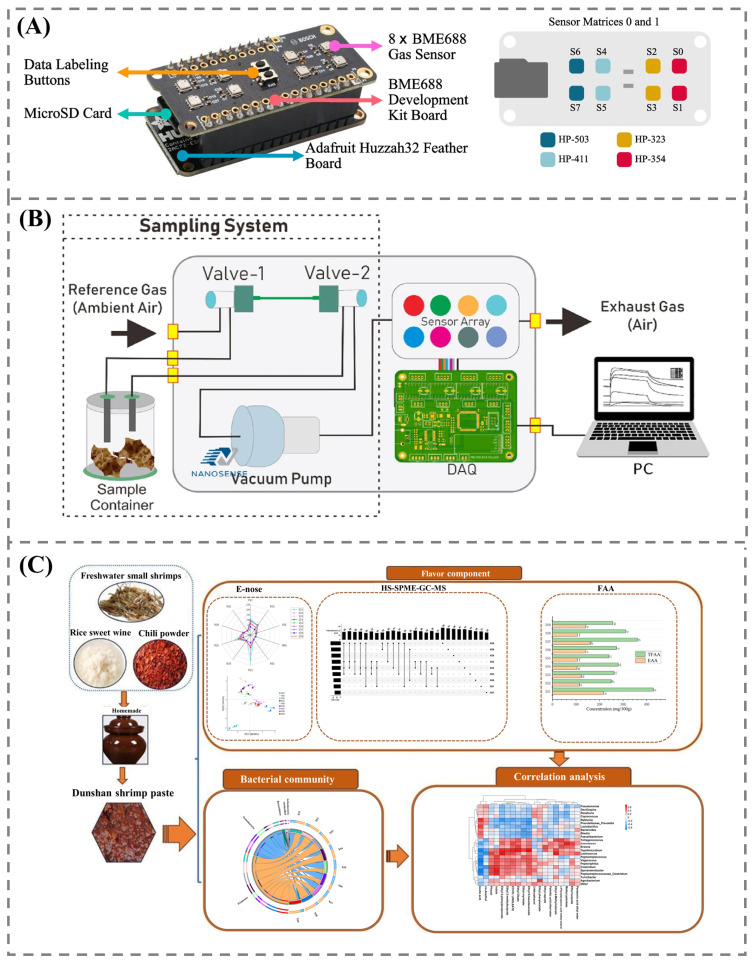
(**A**) BME688 gas sensor developed for detecting olive oil adulteration [48]; (**B**) portable e-nose developed for classifying different types of meat floss [50]; (**C**) e-nose for analysis of the flavor profile of Dushan shrimp paste. Adapted with permission from Ref. [57]. Copyright 2024 Elsevier.

### 3.3. Profiling and Process Optimization

Gas sensors have shown great potential in monitoring tea fermentation by providing an objective assessment of the development of volatile compounds, a task typically carried out by trained sensory personnel. Both multi-sensor and single-sensor systems have demonstrated effectiveness in analyzing the dynamic aroma profiles of tea at different stages of fermentation in black and oolong tea. The results have shown a close correlation with human sensory evaluations. Furthermore, the system identified key response peaks and achieved a classification accuracy of over 80% for aroma variations during low-country Sri Lankan tea fermentation [58]. A more cost-effective approach utilizing a single resistive gas sensor identified fermentation points that correlated with the maximum theaflavin content, which are quality markers recognized by sensory experts [59]. This demonstrates that while multi-sensor systems offer high resolution, a carefully designed single-sensor system can produce reliable quality indicators. A metal-oxide semiconductor array was used to assess the fermentation of oolong tea [60]. The array was able to distinguish between subtle stages of fermentation, with a reduction in grassy notes being identified. This system demonstrated a more sensitive discrimination between intermediate stages compared to black tea aroma profiling.

Apart from being used in tea fermentation, gas sensors have also been applied in various fermentation processes for real-time monitoring and aroma profiling. In sourdough fermentation, a gas sensor array was used to monitor gas emissions, revealing strong correlations with other properties such as pH and total titratable acidity [61]. This provides a cost-effective method for analyzing the fermentation process and controlling it. Li et al. [62] utilized an e-nose to monitor the variations in VOCs during various stages of shrimp paste fermentation, providing valuable insights into the mechanisms of flavor formation (Figure 4A). Similarly, a MOS sensor identified distinct aroma profiles during various stages of dough fermentation and baking [63]. An optimized electronic nose sensor has also been applied in industrial fermentation, where gas emissions during agricultural waste digestion are analyzed, providing real-time operational insight for process optimization [64]. This application highlights the strength of gas sensors in food process optimization and aroma profiling, showcasing their flexibility for different food products and raw materials.

The application of colorimetric gas sensors in the drying of tencha, fermented bean curd, and Baijiu has shown significant progress, moving beyond the detection of major volatile compounds to pattern-based recognition and classification based on complex aroma profiles. Liu et al. [65] integrated a machine learning algorithm (Random Forest) with sensor data obtained during tencha drying to classify the aroma changes that occurred during the drying process (Figure 4B). Similarly, a chemometrics model applied to sensor data obtained during bean curd fermentation enabled both flavor characterization and grading, ensuring quality control throughout the process [66] (Figure 4C). Shui et al. [67] expanded the use of colorimetric sensor arrays by creating a cost-effective chip that was enhanced with nanomaterials. This chip successfully differentiated between various brands of Baijiu, representing a significant advancement in sensor design with the potential to transition sensing applications from the laboratory to real-world settings (Figure 4D).

**Table 2 foods-14-02706-t002:** Gas sensor technologies for food safety and quality monitoring.

**Sensor Type**	**Sample**	**Application**	**Analyte**	**Study Duration**	**Result**	**Reference**
Colorimetric Film	Pork	Quality evaluation	Amine	60 h	Showed distinct color changes at different spoilage stages	[33]
Colorimetric solid-state sensor	Veal, chicken, fish	Spoilage detection	Ammonia	5 days	Developed sensor with LOD of 0.02 ppm	[34]
Colorimetric sensor array	Wheat	Spoilage detection	VOCs		Mold infection in wheat	[68]
Colorimetric sensor array	Chilled beef	Freshness monitoring	Trimethylamine	18 days	Developed sensor with LOD of 8.02 ppb	[35]
Colorimetric sensor array	Rice	Freshness evaluation	Alcohols, aldehydes, alkenes, alkanes, ketones, organic acid, heterocyclic compounds	-	Discrimination of aged and fresh rice (100%)	[69]
Colorimetric Film	Milk, fish	Spoilage detection	VOCs	9 days	Biomaterial based edible and pH-sensitive film	[70]
Colorimetric Film	Pork, chicken, salmon, and shrimp	Spoilage detection	Ammonia, dimethylamine, and trimethylamine	7 days	LODs were determined to be 0.26 μM for NH_3_, 0.24 μM for DMA, and 0.38 μM for TMA	[71]
Colorimetric Film	Beef	Spoilage detection	Ammonia	8 days	Developed a photothermally stable and ammonia-responsive film	[72]
(001)TiO_2_/MXene sensor	Fish, pork and shrimp	Quality monitoring	Ammonia	36 h	Developed sensor with LOD of 156 ppt	[36]
NiCo_2_O_4_-ZnO sensor	-	Quality evaluation	Trimethylamine	-	Improved response value	[37]
Electrochemical, infrared (IR), MOS	Lamp	Quality evaluation	O_2_, CO_2_, and NH_3_		Gas composition analysis along with impedance	[38]
Electrochemical sensor	Pork	Freshness monitoring	Trimethylamine		Range: 3.33 μg/L–1200 μg/L	[73]
E-nose, fluorescence hyperspectral imaging	Pork	Freshness monitoring	Alcohols, aldehydes, ketones, alkanes, and sulfides	7 days	End-to-end data fusion approach for freshness	[74]
E-nose	Spinach	Quality evaluation during storage	Hydrogen sulfide, methane, alcohol, ammonia, and carbon monoxide	12 days	Optimized sensor array for odors classification	[40]
Color-sensitive gas sensor array	Wheat flour	Quality evaluation during storage	VOCs	6 months	Odor information of flour samples of different storage periods	[75]
E-nose	Edible oil	Quality evaluation during storage	Hydrogen, carbon monoxide, methane, ethanol, toluene, acetone, and formaldehyde	5 days	Change in quality during storage	[76]
E-nose	Green tea	Quality evaluation	Alcohols, aldehydes, and esters	-	Improved classification accuracy	[43]
E-nose	Apple	Spoilage monitoring	Nitrogen oxide, ammonia, hydrogen, alkanes, sulfides, alcohol, aromatic compounds, inorganic sulfur, organic compounds, methane, and aliphatic organic compounds	7 days	Integrated terminal and remote platform enabled real-time monitoring	[77]
E-nose	Egg	Sensory quality traits evaluation	Ammonia	42 days	Maturity level recognition at 95% accuracy	[44]
E-nose	Chicken drumstick	Quality evaluation	Alcohols, aldehydes, phenols, ketones, and cis-anethol	-	Optimum sugar smoking effects on flavors	[39]
Colorimetric film	Apple	Spoilage moni-toring	CO_2_	5 days	High recognition rate for spoilage	[78]
E-nose	Chilled Chicken	Quality evaluation	Sulfides, organic sulfides, and hydrides	3 days	Difference in VOC produced by different species of bacteria	[79]
Gas sensor array	Apple	Quality of pathogen-contaminated apples	VOCs	7 days	Prototype for early warning of apple spoilage	[80]
E-nose, Colorimetric sensor array	Fermented bean curd	Flavor quality analysis	VOCs	-	Determine ripeness and predict hardness	[81]
BME688 sensor	Olive oil	Adulteration detection	VOCs, carbon monoxide, and hydrogen	-	High sensitive detection of sunflower oil adulteration	[48]
E-nose	Powdered milk	Adulteration detection	2-propanone, 5-methyl-2(3H) furanone	-	Whey adulteration in powdered milk	[47]
E-nose	Minced chicken meat	Adulteration detection	Alcohols, aldehydes, ketones, aromatics, and other organic vapors	-	Detection of soybean protein isolate adulteration	[49]
E-nose, GC-MS	Sesame oil	Adulteration detection	Alcohol, organic solvents, SO_2_, CO, alkenes, ammonia, benzene, sulfides, hydrogen, methane	-	Adulteration with soybean and corn oils	[46]
E-nose	Beef	Adulteration detection	VOCs	-	Adulteration with pork	[82]
E-nose	*Zanthoxylum bungeanum Maxim*	Discrimination based on geographical origin	VOCs	-	Discrimination based on geographical origin	[51]
E-nose and e-tongue	Red wine	Discrimination based on geographical origin	Alcohols, esters, aldehydes, and ketones	-	Discrimination based on geographical origins, brands, and grape varieties	[52]
Smartphone-based colorimetric sensor array	Rice	Discrimination based on geographical origin	VOCs	-	Discrimination based on geographical origin	[53]
Colorimetric sensor array	Edible bird’s nests	Discrimination based on geographical origin	Octadecanoic acid, propanetriol, and 4-terpenol	-	Discrimination based on geographical origin	[83]
E-nose	Meat floss	Classification	Hydrocarbons and alcohols	-	Differentiate beef, chicken, and pork meat floss	[50]
E-nose	Lemon juice	Quality evaluation	VOCs	120 days	Freshness during storage	[84]
Proton-Transfer-Reaction Mass Spectrometry (PTR-MS)	Occidental pears	Fruit ripeness monitoring	VOCs (esters and terpenes)	-	Identification of three ripening stages of occidental pears	[85]
E-nose, e-tongue	Fermented soybean paste	Flavor quality analysis	VOCs	-	Evaluation of sensory properties and overall flavor quality	[55]
E-nose, e-tongue, GC-MS	Soybean paste	Flavor quality analysis	Nitrogen oxides, ammonia, hydrogen, methane, H2S, terpenes, alcohol, alkenes, and aromatic organic compounds	-	Evaluation of sensory properties and overall flavor quality	[56]
E-nose	White tea	Authentication	Alcohols, esters, aldehydes, ketones, alkenes, hydrocarbons, ammonia, and alkyl aromatic compounds	-	Vintage authentication by integrating appearance, taste and aroma assessments.	[86]
E-nose	Tea	Monitor fermentation	Isobutane, propane, methane, hydrogen, smoke, benzene, hydrogen, and alcohol	-	Detection of fermentation stages and detects aroma changes	[58]
MOS sensor	Oolong tea	Monitor fermentation	Air contaminants, odorous gases, hydrocarbons, solvents, and sulfur compounds	-	Control flavor quality during manufacturing	[60]
Generic resistive gas sensor	Black tea	Monitor fermentation	VOCs	-	Optimizes fermentation time	[59]
MOS sensor	Oolong tea	Monitor oxidation	Ammonia, hydrogen, ethanol, sulfides, benzene, methane, propane, butane, alkenes, toluene, acetone, ethanol, and formaldehyde	-	Monitor oxidation process	[87]
E-nose, e-tongue	*Tremella aurantialba*	Monitor and detect the fermentation process	Methane, ethane, dimethyl methane, hydrogen sulfide, and alcohol	-	Predict key chemical indicators	[88]
Gas sensor array	Sourdough	Monitor fermentation	Oxygen, carbon dioxide, and hydrogen sulfides	-	Online gas measurements predict pH and acidity	[61]
Headspace-gas chromatography-ion mobility spectrometry	Shrimp paste samples	Monitor fermentation	VOCs	-	Alcohols and amines dominated volatile compounds	[62]
E-nose	Bread	Monitor fermentation	Alcohols, aldehydes, esters, ketones, terpenoids, pyridines, hydrocarbons, and amides	-	Pyridines are characteristic for emissions during baking	[63]
E-nose	Rice and wheat crop residues	Understand bioethanol production dynamics	Ammonia, NO_2_, i-butane, propane, methane, alcohol, hydrogen, CO, toluene, and xylene	-	Artificial intelligence optimized sensor responses for classification and prediction	[64]
E-nose	Mulberry wine	Monitor fermentation	Alcohol, H_2_S, terpenes, organic sulfur, nitrogen, oxygen, ammonia, alkenes, and methane	-	Effect of selenium-enriched yeast fermentation on flavor profiles	[89]
E-nose, GC-MS	Steam bread	Aroma assessment	Alcohols, esters, aldehydes, and furan	-	Effects of multi-strain co-fermentation flavor profiles	[90]
Colorimetric sensor array	Tencha	Aroma assessment	VOCs	-	Developed an olfactory visualization system to optimize drying	[65]
Gas chromatography-ion mobility spectrometry (GC-IMS) and gas chromatography-mass spectrometry-olfactometry (GC-MS-O) techniques.	Sturgeon meat	Flavor stability analysis	VOCs	-	Identify optimal steaming conditions and formic acid as crucial volatile compounds contribute flavor	[91]
Colorimetric sensor array	Fermented bean curd	Flavor quality analysis	VOCs	-	Discrimination based on different brand	[66]
MQ-3 gas sensor	Glucose	Quantitatively assesses fermentation	Alcohol	-	Identified correlation between gas bubble formation and alcohol production	[92]
Colorimetric sensor array	Baijiu	Quality control	Ethyl caproate, ethyl lactate, n-propanol, n-butanol, isobutanol, isoamyl alcohol, acetic acid, butyric acid, and capric acid	-	Discrimination brands and authenticity	[67]
Colorimetric sensor array	Baijiu	Quality evaluation	VOCs	-	Discrimination of different grades	[93]
E-nose	Coffee leaves	Quality control	Nitrogen oxides, short-chain alkanes, sulfur-inorganic compounds, alcohols, aldehydes, ketones, and sulfur-containing organic compounds	-	Difference in aroma profiles of freeze-dried and hot-air dried leaves	[94]

**Figure 4 foods-14-02706-f004:**
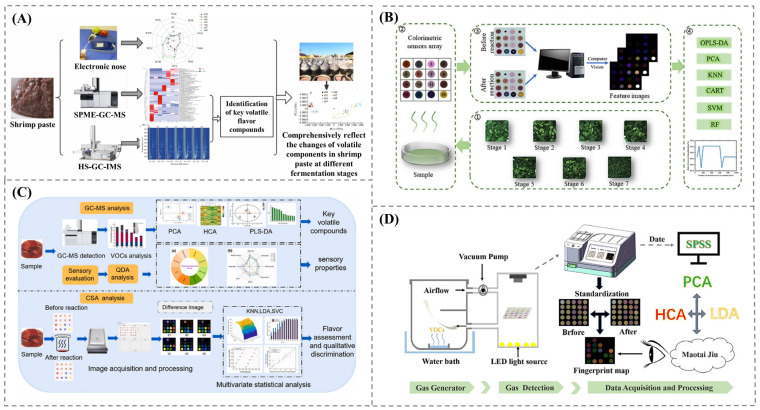
(**A**) E-nose monitoring variations in VOCs during various stages of shrimp paste fermentation. Adapted with permission from Ref. [62]. Copyright 2022 Elsevier. (**B**) colorimetric gas sensors applied to tencha drying to classify the aroma changes that occurred during the drying process. Adapted with permission from Ref. [65]. Copyright 2024 Elsevier. (**C**) colorimetric sensor array applied during bean curd fermentation for flavor characterization and grading. Adapted with permission from Ref. [66]. Copyright 2024 Elsevier. (**D**) colorimetric sensor arrays used for differentiation of various brands of Baijiu. Adapted with permission from Ref. [67]. Copyright 2024 Elsevier.

## 4. Data Acquisition and Pattern Recognition

A reliable data analysis method is essential for analyzing the data collected from gas sensors in order to draw meaningful conclusions. This is crucial for the decision-making process in food safety and quality analysis. It is particularly important in the food industry because of the intricate complexity of the volatile compounds produced by food products, which makes it challenging to extract information directly from raw data [95]. Furthermore, gas sensors face serious challenges such as selectivity, limit of detection, and response time. Proper data analysis and processing techniques are necessary to effectively address these issues, ensuring better performance of the sensor in real-world environments. Data analysis has been shown to significantly improve the performance of chemiresistive gas sensors by enhancing the monitoring and optimization of their response to gas components [96].

Machine learning models improve sensor performance by enhancing calibration methods, adapting to environmental variability, and enabling trace-level detection. Gaussian Process Regression (GPR) was used to calibrate a gas sensor by considering the environmental factors that influence metal-oxide sensors, leading to improved methane detection accuracy [97]. The study demonstrated that the efficacy of GPR models can be optimized by tuning their kernels with hyper-parameters, a task that is not feasible with ANN models. Ng et al. [98] utilized a combination of Multi-Layer Perceptron (MLP) and Long Short-Term Memory (LSTM) models to effectively compensate for variations in humidity and temperature in the environment using gas sensor data. The incorporation of machine learning has improved the accuracy of the sensor in detecting CO, NO_2_, and C_6_H_6_.

Machine learning and deep learning techniques have proven their effectiveness in feature extraction and pattern recognition in gas sensor data. They have the ability to analyze complex data and recognize subtle patterns, allowing for the sensor to distinguish between similar gases. Lee et al. [99] trained a 1D CNN on the unique patterns of different gases, including acetone, ammonia, ethanol, and nitrogen dioxide, under varying temperature conditions without separate feature extraction (Figure 5A). This resulted in an enhanced ability to identify the gases. This enabled the model to learn the specific characteristics of spice aromas and achieve a classification accuracy of 96.1% for coriander, cilantro, star anise, and licorice. The implementation of a 1D CNN was directly applied to raw sensor data without additional feature extraction, simplifying the entire data processing method. The Support Vector Machine (SVM) excels at pattern recognition by selecting an optimal hyperplane that separates different classes in the gas sensor data. The SVM successfully classified lemon juice samples into five categories based on their quality during the storage period [84].

It also promotes a more holistic approach that integrates the entire process of sensing and monitoring, paving the way for streamlined and automated intelligent decision-making. The integrated framework improves operational efficiency and accuracy by leveraging real-time data analytics. By incorporating near-field communication (NFC) technology, the sensors are able to transfer data wirelessly to smartphones, resulting in quick decision-making during food spoilage, as shown in Figure 5B,C [100]. This wireless data transfer facilitates streamlining the monitoring process, benefiting both consumers and suppliers. Furthermore, the use of wireless sensor networks, which consist of multiple sensor nodes that continuously collect and transmit data to a central monitoring system, ensures timely intervention during food storage [101].

The integration of suitable gas sensors into the IoT significantly enhances the capability and widens the applications for food safety and quality analysis [102]. The IoT-enabled gas sensor could continuously monitor VOCs through real-time data transmission, providing early warning of spoilage continuously. For beef quality assessment, Damdam et al. [103] utilized an IoT-based electronic nose system to detect spoilage markers such as CO_2_, ammonia, and ethylene, thereby eliminating the need for conventional microbial tests. In contrast, Goyal et al. [104] integrated IoT technology with a multiple sensor array to monitor VOCs, pH, and temperature simultaneously, enabling the detection of adulteration in milk. Figure 5D shows the data acquisition system employed for seamless transmission of data to cloud server. This demonstrates that IoT integration facilitates real-time data acquisition and remote monitoring, as well as customized solutions based on the specific requirements of the application and food matrix.

The integration of advanced data acquisition methods and pattern recognition techniques is of significant importance for developing gas sensors for food safety and quality evaluation. These advanced techniques enable more accurate and reliable detection of spoilage, adulteration, and contamination, while addressing the limitations associated with conventional processing methods. Integrative approaches facilitate intelligent and automated decision-making processes, enhancing the versatility of gas sensors for use in various environments.

**Figure 5 foods-14-02706-f005:**
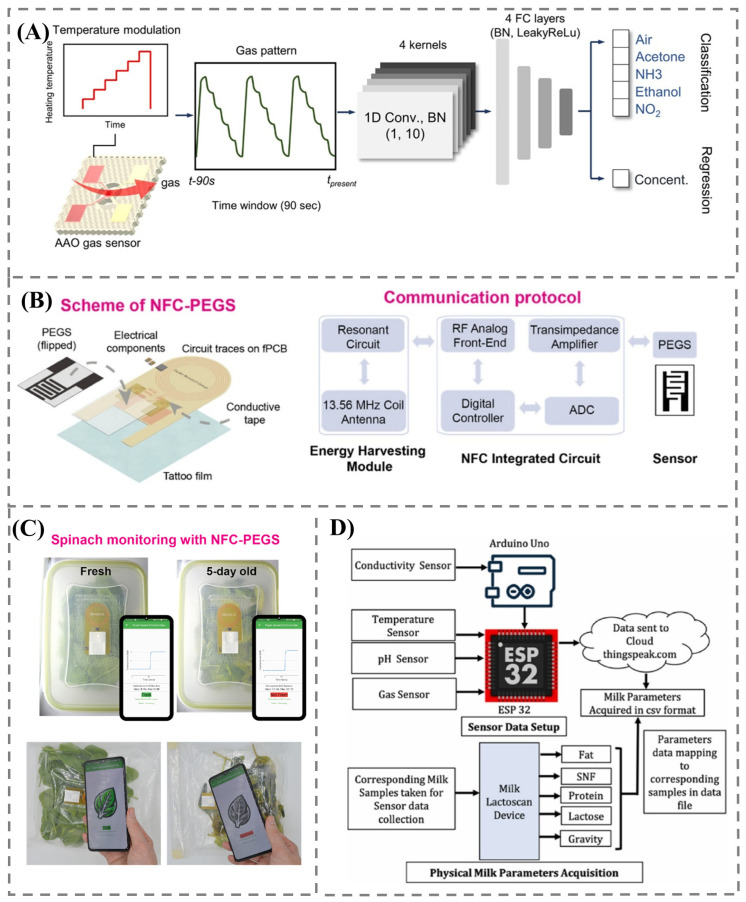
(**A**) The 1D CNN-based multigas classification using the temperature-modulated operation of the gas sensor. Adapted with permission from Ref. [99]. Copyright 2025 American Chemical Society. (**B**) NFC-powered sensing device developed for wireless monitoring of food freshness [100]. (**C**) NFC-powered gas sensor integrated into spinach packaging and interfaced with a smartphone application [100]. (**D**) Data acquisition system employed for the transmission of data to a cloud server, promoting real-time monitoring. Adapted with permission from Ref. [104]. Copyright 2024 Elsevier.

## 5. Challenges and Future Trends

Gas sensors have a wide range of applications in the evaluation of food safety and quality, including, but not limited to, indicating freshness and detecting adulteration. The VOC profile of a food product is frequently linked to its physiochemical characteristics, allowing for the prediction of consumer preferences. The multisensor array system is capable of capturing the additional properties of food in conjunction with VOCs, providing valuable insight into the interaction of various parameters that contribute to quality changes. The previous section discusses the primary applications and advancements of gas sensors. However, there are still limitations around utilizing gas sensors for food analysis:Gas sensors exhibit low selectivity, making it challenging to accurately identify specific components for detection. Selectivity is a crucial factor in the effectiveness of gas sensors, as different food products exhibit distinct characteristic aroma profiles. The presence of VOCs with similar molecular structures may hinder the ability of basic gas sensors to accurately detect specific target compounds, resulting in false results. Therefore, it is crucial to focus on the development of gas sensors with high specificity in order to improve detection accuracy. Utilizing novel sensing materials, specifically metal-oxide nanoparticles, has been found to be an effective method for addressing the selectivity issue in gas sensors. The incorporation of these materials into mesoporous structures has been shown to enhance gas absorption and consequently increase sensor sensitivity, ultimately improving sensor selectivity.The integration of gas sensors into food packaging or processing lines is a complex task due to the rigid nature of most sensors which are not suitable for packaging systems or processing lines. It is challenging to incorporate sensors into various packaging materials and food containers without compromising the integrity of the system. Therefore, the development of flexible gas sensors is crucial for the successful integration of sensors into complete packaging systems.Certain sensors, such as MOS and surface acoustic wave sensors, have limitations in their application for the real-time analysis of food products, particularly perishable foods, due to their long recovery time. It is imperative to obtain information quickly in order to implement measures to prevent and address issues such as spoilage, adulteration, and changes in quality. The utilization of nanostructures as a sensitive material, as well as doping and composite material sensors, has been shown to be highly effective in reducing operation time and improving absorption and desorption kinetics. Incorporating a MEMS platform or layer onto an existing gas sensor is a crucial method for achieving faster cycling between the response and recovery phases.A portable detection device utilizing a wireless communication protocol is essential for the continuous monitoring of food freshness within the supply chain management system. This real-time monitoring will decrease the need for manual sampling and laboratory testing, while also aiding in the prediction of the maintenance required to ensure food quality and safety. To optimize supply chain management for perishable food over an extended period, low-power sensors are utilized in conjunction with IoT nodes to integrate data on temperature, humidity, and CO_2_ levels. This comprehensive dataset provides insight into the environmental conditions affecting food products, enabling informed decision-making based on the entirety of available information.Gas sensor readings are significantly influenced by environmental factors, particularly temperature and humidity. Unpredictable fluctuations in these factors can lead to unreliable readings. This attribute is of great importance for a gas sensor, particularly when utilized in monitoring food products in fluctuating storage environments. In the future, it is recommended to implement a system that incorporates humidity and temperature compensations to enhance the robustness of the sensor and enable its operation in diverse environmental conditions.The absence of a universal database for evaluating the VOCs associated with food spoilage hinders the utilization of gas sensors. In the future, there is a need for the establishment of a centralized Open Access database for referencing compounds detected using gases in order to facilitate reliable decision-making.

Gas sensors show potential in the field of food safety and quality assessment; however, there exist numerous challenges that need to be overcome to fully exploit the capabilities of this technology. The optimization of sensor selectivity, reductions in response and recovery time, enhancements in environmental robustness, and the facilitation of their integration into flexible packaging systems are critical areas for improvement. The integration of innovative sensing materials and nanostructures, in conjunction with the implementation of IoT techniques and remote monitoring, presents a promising strategy for overcoming current limitations. Future research should prioritize the development of adaptable, intelligent, and portable sensor systems that can seamlessly integrate into supply chain management systems to enhance food quality analysis options.

## 6. Conclusions

Gas sensors have emerged as a significant component in the development of food safety and quality evaluation methods. The VOCs detected by the gas sensors directly correlate with the quality parameters associated with food spoilage and degradation. The non-destructive nature of gas sensing methods offers a real-time solution for assessing food quality throughout the supply chain. The detected VOCs can be linked to other physicochemical properties of the food to gain a comprehensive understanding of the changes occurring in food products during processing. Additionally, an integrated approach that incorporates various sensors covering all aspects of food through a sensor array system and data integration provides a holistic approach for more accurate assessments. A diverse range of gas sensor technologies, from single MOS sensors to advanced sensor arrays, offer rapid, sensitive, and real-time detection of VOCs in food products. While MOS, electrochemical, and conducting polymer sensors operate based on electrical signal transduction mechanisms, colorimetric and fluorescence sensors provide visual indicators, making them more suitable for consumer-friendly monitoring without the need for additional equipment. Furthermore, the utilization of a sensor array comprising multiple sensor systems facilitates the emulation of human sensory perception, thereby allowing for a more comprehensive analysis of food properties. Technological advancements and innovative approaches provide a solution to address the challenges posed by environmental variability, sensor selectivity and sensitivity, response time, data processing complexity, cost, and accessibility. Implementing machine learning and deep learning, developing an environmental compensation system, and utilizing advanced sensor materials are the focus of research for optimizing performance and expanding applications in the future. Furthermore, the combination of gas sensors with sophisticated data acquisition methods, pattern recognition techniques, and the IoT has enabled the development of intelligent monitoring systems that can offer real-time information on food quality.

## Figures and Tables

**Table 1 foods-14-02706-t001:** Comparative overview of gas-sensing technologies.

Technique	Key Characteristics	Fabrication Method	Advantages	Limitations
MOS Sensor	Detect gases by measuring changes in electrical resistance due to reversible interactions between gases and the metal-oxide surface	Wafer fabrication, oxidation, mask generation, photolithography, diffusion, and deposition	High gas response, reversible reactions, cost-effective, sensitive to freshness marker gases, operable across a range of temperatures, and compatible with sensor array integration	Limited selectivity, slow response and recovery times, and limited mass transfer in the gas phase
Electrochemical sensor	Convert chemical concentrations into electrical signals via redox reactions, enabling selective and accurate gas detection	Electrochemical deposition, electroless deposition, microspotting, dip-pen lithography, and self-assembly	Highly sensitive, selective, rapid response, portable, and adaptable to various conditions, with compatibility for sensor array integration	Cross-sensitivity to various gases, limited lifespan, and sensitivity to temperature and humidity
Optical sensor	Detect gases through chemical reactions between the gas and a chromogenic dye, which results in absorbance or fluorescence shifts.	Dip-coating technique, electrospun nanofibers, electrochemical writing, inkjet printing, and sol–gel techniques	Simple, cost-effective, and provides visual results for gas detection	Environmental interference, poor long-term stability, single-use design, rapid consumption of sensing materials, and challenges in calibration
Conducting Polymer Sensor	Detect gases through chemiresistive behavior, where electrical resistance changes upon gas exposure.	Sol–gel, in situ oxidative polymerization, template-based methods, solid-state synthesis, and oxidative chemical vapor deposition	Large-scale production, tunable electrical properties, flexibility, biocompatibility, ease of fabrication, and high sensitivity to gases like ammonia and hydrogen sulfide	Poor long-term stability, environmental interference, potential irreversible changes upon gas exposure, and high production costs
Sensor Array	Detect multiple VOCs simultaneously, mimicking the human olfactory system through a combination of diverse gas sensors.	Fabrication of patterned devices using engineered nanomaterials and integration of sensors into array systems via micro-electro-mechanical systems (MEMS)	Rapid detection, stability, portability, compactness, ability to identify complex gas mixtures, and adaptable for multiple applications	Response time issues, partial sensor specificity, interference from overlapping compounds, and need for advanced pattern recognition for accurate odor classification

## Data Availability

No new data were created or analyzed in this study. Data sharing is not applicable to this article.

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
