# Peer review of "Intelligent Gas Sensors for Food Safety and Quality Monitoring: Advances, Applications, and Future Directions"

_foods, 2025, doi:10.3390/foods14152706_

Round 1

Reviewer 1 Report

Comments and Suggestions for Authors
  • It is better to state the type of gases released from the food for Table 1, instead of just stating VOC because the type of VOC gases is huge. Please also add a parameter such as the time taken for the food to be spoiled. It can be used as guidance for other researchers to conduct their research.
  • To add a topic about the classification method used to identify the type of gas, especially by using an e-nose since the title of this paper is intelligent. therefore, the AI method should be discussed in the study.

Author Response

Comment 1: It is better to state the type of gases released from the food for Table 1, instead of just stating VOC because the type of VOC gases is huge. Please also add a parameter such as the time taken for the food to be spoiled. It can be used as guidance for other researchers to conduct their research.

To add a topic about the classification method used to identify the type of gas, especially by using an e-nose since the title of this paper is intelligent. therefore, the AI method should be discussed in the study.

Response: Thank you for the suggestion. Table 2 has been updated to include specific volatile compounds detected in each study, rather than referring to them generally as VOCs. A separate column indicating the total time until food spoilage was also added, where such information was available. Please note that this parameter is only applicable to studies specifically focused on spoilage analysis.

 The classification methods, including those involving electronic noses and AI based approaches, are already discussed in Section 4 (Data Acquisition and Pattern Recognition) to reflect their relevance to intelligent sensing systems.

Reviewer 2 Report

Comments and Suggestions for Authors

 This review article provides a comprehensive and well-structured overview of various gas sensor technologies, including metal oxide semiconductor (MOS) sensors, electrochemical sensors, optical sensors, conducting polymer sensors, and sensor arrays. It effectively highlights their diverse applications in ensuring food safety and quality. The authors’ attention to emerging technologies—such as smart packaging, the Internet of Things (IoT), machine learning, and deep learning—reflects a modern and practical approach. Moreover, the discussion on critical challenges such as selectivity, response time, long-term stability, and sensor integration into packaging systems demonstrates a forward-looking perspective. By presenting a wide range of real-world examples, including the detection of spoilage, freshness, adulteration, and authenticity of food products, the article clearly illustrates the practical potential of these sensing technologies.

Overall, this is a well-prepared and timely review. However, I would like to offer the following suggestions to further enhance the quality and usefulness of the manuscript:

  1. Consider including a comparative table that summarizes the key characteristics, advantages, limitations, and specific applications of each gas sensing technology discussed.
  2. Provide a multi-criteria comparison of the sensor groups, considering aspects such as cost-effectiveness, ease of use, detection accuracy, sensitivity, and scalability.
  3. Include a brief discussion on the technological evolution of each sensor type—from its initial development to its current state or commercial/industrial implementation.

Author Response

General comment: This review article provides a comprehensive and well-structured overview of various gas sensor technologies, including metal oxide semiconductor (MOS) sensors, electrochemical sensors, optical sensors, conducting polymer sensors, and sensor arrays. It effectively highlights their diverse applications in ensuring food safety and quality. The authors’ attention to emerging technologies—such as smart packaging, the Internet of Things (IoT), machine learning, and deep learning—reflects a modern and practical approach. Moreover, the discussion on critical challenges such as selectivity, response time, long-term stability, and sensor integration into packaging systems demonstrates a forward-looking perspective. By presenting a wide range of real-world examples, including the detection of spoilage, freshness, adulteration, and authenticity of food products, the article clearly illustrates the practical potential of these sensing technologies.

Overall, this is a well-prepared and timely review. However, I would like to offer the following suggestions to further enhance the quality and usefulness of the manuscript:

Comment 1:Consider including a comparative table that summarizes the key characteristics, advantages, limitations, and specific applications of each gas sensing technology discussed.

Response: Thank you for the suggestion. A comparative table (Table 1) has been added to the manuscript, summarizing the key characteristics, advantages, limitations, and manufacturing methods of each gas sensing technology discussed.

Comment 2: Provide a multi-criteria comparison of the sensor groups, considering aspects such as cost-effectiveness, ease of use, detection accuracy, sensitivity, and scalability.

Response: Thank you for the helpful suggestion. A multi-criteria comparison of the sensor types, highlighting their key characteristics, advantages, and limitations, has been added as Table 1.

Comment 3: Include a brief discussion on the technological evolution of each sensor type—from its initial development to its current state or commercial/industrial implementation.

Response: Thank you for the suggestion. A brief discussion on the technological evolution of each sensor type has been added to the manuscript in corresponding section of the sensors.